# Tako-Tsubo Syndrome in Amyotrophic Lateral Sclerosis: Single-Center Case Series and Brief Literature Review

**DOI:** 10.3390/ijms241512096

**Published:** 2023-07-28

**Authors:** Giovanni Napoli, Martina Rubin, Gianni Cutillo, Paride Schito, Tommaso Russo, Angelo Quattrini, Massimo Filippi, Nilo Riva

**Affiliations:** 1Neurorehabilitation, Neurology Unit and Neurophysiology Unit, San Raffaele Scientific Institute, 20132 Milan, Italy; napoli.giovanni@hsr.it (G.N.); rubin.martina@hsr.it (M.R.); cutillo.gianni@hsr.it (G.C.); schito.paride@hsr.it (P.S.); russo.tommaso@hsr.it (T.R.); filippi.massimo@hsr.it (M.F.); 2Neuroimaging Research Unit, Institute of Experimental Neurology, Division of Neuroscience, San Raffaele Scientific Institute, 20132 Milan, Italy; 3Experimental Neuropathology Unit, Institute of Experimental Neurology (INSPE), Division of Neuroscience, San Raffaele Scientific Institute, 20132 Milan, Italy; quattrini.angelo@hsr.it; 4Vita-Salute San Raffaele University, 20132 Milan, Italy

**Keywords:** motor neuron disease, autonomic system, mortality, non-invasive ventilation, PEG, NIMV, vascular, cardio-vascular

## Abstract

Amyotrophic lateral sclerosis (ALS) is a neurodegenerative disease with variable phenotypic expressions which has been associated with autonomic dysfunction. The cardiovascular system seems to be affected especially in the context of bulbar involvement. We describe four new cases of Tako-Tsubo syndrome (TTS) in ALS patients with an appraisal of the literature. We present a late-stage ALS patient with prominent bulbar involvement that presented TTS during hospitalization. We then retrospectively identify three additional ALS–TTS cases reporting relevant clinical findings. TTS cardiomyopathy has been observed in different acute neurological conditions, and the co-occurrence of ALS and TTS has already been reported. Cardiovascular autonomic dysfunctions have been described in ALS, especially in the context of an advanced diseases and with bulbar involvement. Noradrenergic hyperfunction linked to sympathetic denervation and ventilatory deficits coupled in different instances with a trigger event could play a synergistic role in the development of TTS in ALS. Sympathetic hyperfunctioning and ventilatory deficits in conjunction with cardiac autonomic nerves impairment may play a role in the development of TTS in a context of ALS.

## 1. Introduction

ALS is a fatal neurodegenerative disease that generally presents during the fifth to sixth decades, with concomitant upper motor neuron (UMN) and lower motor neuron (LMN) signs [1,2]. It causes progressive paralysis leading to respiratory failure within 3–5 years. In about 20% of cases, ALS presents with a bulbar onset, which is generally associated with an earlier development of nutritional and/or respiratory failure and, consequently, a poorer prognosis. However, up to 80% of ALS patients develop bulbar involvement regardless of the onset site [1,3]. While ALS has traditionally been considered a pure motor disease, recent advances in our understanding have revealed heterogeneous extra-motor involvement [2]. In addition to the well-recognized cognitive impairment, ALS is known to be associated with varying degrees of autonomic and cardiovascular dysfunction. These dysfunctions typically present during the advanced stages of the disease and are often associated with bulbar involvement [4,5,6,7,8]. However, they have also been described in early or even prodromal stages [5,9]. ALS has been linked to Tako-Tsubo syndrome (TTS) [7,10,11,12,13], an acute cardiomyopathy triggered by stress-related adrenergic dysregulation, mimicking acute coronary syndrome. TTS causes apical ballooning and transient left ventricular dysfunction. Although TTS is generally considered reversible, affected patients may experience persistent cardiac dysfunction even after the acute phase and the resolution of apical ballooning [14,15]. In this report, we present four new cases of ALS–TTS co-occurrence, focusing on autonomic dysfunction. We also provide an overview of the existing literature and speculate on the possible underlying pathophysiological mechanisms.

## 2. Case Presentation

A 59-year-old man with a 6-year history of ALS, diagnosed according to the revised El Escorial criteria [16] and the Awaji criteria [17], was admitted to the Neurorehabilitation Unit due to progressive development of dysphagia and respiratory failure in order to assess the need for non-invasive ventilation (NIV) or percutaneous endoscopic gastrostomy (PEG) placement.

At the time of diagnosis, the patient presented with a spinal onset, initially involving the lower limbs, and riluzole therapy was started. Two years later, he required a walker for ambulation, and three years later, he became wheelchair-bound. The upper limbs were gradually affected, with relatively preserved distal function.

In the two months leading up to admission, the patient reported progressive dysphagia, dyspnea with minimal effort or prolonged speech, orthopnea, and ineffective cough. One month prior, he underwent a cardiological assessment due to new-onset palpitations, and sporadic paroxysmal supraventricular tachycardia runs were observed on an electrocardiogram (ECG). Consequently, he was started on Flecainide. No other cardiac comorbidities were reported.

Upon admission, a routine ECG revealed a prolonged cQT interval (>530 ms), a right bundle branch block, and an anterior left hemiblock with T wave alterations suggestive of possible cardiac ischemia (Figure 1A). However, the patient was asymptomatic. Treatment with intravenous acetylsalicylate was initiated and later switched to a low-dose oral formulation, along with metoprolol and ramipril. Blood tests showed a mild elevation of troponin T and a severe elevation of pro-BNP. Echocardiography revealed concentric thickening of the left ventricle with a 45% ejection fraction (FE) and medial-apical dyskinesia, which was suggestive of Tako-Tsubo syndrome (TTS). To confirm the suspicion of TTS, a coronary CT scan was performed, which showed no stenosis in the coronary arteries. As a result, the previous therapy was discontinued, and Bisoprolol was initiated. In the following days, Troponin T and pro-BNP levels, as well as ECG findings (Figure 1B) and echocardiography results, progressively normalized.

### 2.1. Case Series

Reviewing a 10-year cohort of approximately 800 ALS patients, we retrospectively identified three additional cases of ALS–TTS (Table 1). The onset of TTS occurred at different disease durations, ranging from 38 to 73 months. Three out of the four patients had a spinal onset, with two presenting classical upper and lower motor neuron (UMN and LMN) signs, and one showing isolated LMN involvement. The fourth patient (pt.2) had a bulbar onset, characterized by dysarthria. All patients received treatment with riluzole and underwent testing for ALS-related genes, including a panel of 30 genes such as SOD1, TARDBP1, and FUS, as well as repeat-primed PCR for detecting the C9orf72 repeat expansion [18]. A C9orf72 mutation was detected only in the patient with bulbar onset, while no relevant mutations were found in the others. At the time of TTS diagnosis, two patients had a Revised Amyotrophic Lateral Sclerosis Functional Rating Scale score below 20, indicating functional impairment. Patient 2 (pt.2) had mild bulbar symptoms, including mild drooling, slight dysarthria, and dysphagia. All patients exhibited apical dyskinesia on echocardiography and elevated levels of Troponin T. ECG findings showed non-specific changes (Table 1). Three patients underwent coronary CT or angiography, which showed no evidence of coronary artery obstruction. Three patients had mild cardiovascular comorbidities, such as supraventricular tachycardia and hypertension, while patient 4 (pt.4) had coronary artery disease. Remarkably, pt.2, despite the bulbar onset, developed TTS without concomitant respiratory or nutritional failure. In contrast, pt.1 and pt.3 developed TTS concurrently with non-invasive ventilation (NIV) initiation, and pt.4 experienced TTS coinciding with tracheostomy and percutaneous endoscopic gastrostomy (PEG) placement. A potential trigger event for TTS was identified in pt.2 and pt.3, who were diagnosed with pulmonary embolism (PE) and a urinary tract infection, respectively. Pt.4 experienced TTS in association with tracheostomy and PEG placement, while in pt.1, TTS was an incidental finding.

### 2.2. Literature Review

The literature exploring the ALS–TTS association is heterogenous, and few reports could be retrieved. In our review, we found 10 relevant articles reporting on TTS in ALS, comprising a total of 32 patients (Table 2). In two cases [11,19], TTS was presented in association with overt bulbar signs, while Choi [20] reported nine patients with cervical or bulbar onset which developed TTS in late ALS stages. Data on respiratory failure were available for 13 patients. A TTS putative trigger was reported in all cases except for a single series [21].

## 3. Discussion

In our cohort, three ALS patients were diagnosed with TTS at relatively advanced stages, with increasingly evident bulbar impairment. Interestingly, the only patient with a bulbar onset developed TTS in an earlier disease stage, with only modest bulbar symptoms, but exhibited the most severe cardiac involvement. ALS has been associated with varying degrees of autonomic dysfunction. Parasympathetic hypofunctioning has been linked to altered salivary and gastrointestinal excretion, orthostatic or nocturnal hypotension, and decreased heart rate variability (HRV) [5,6]. On the other hand, sympathetic hyperfunctioning is associated with an increased risk of cardiac arrest [6,21,25]. While dysautonomia in ALS typically appears in late stages, it has rarely been reported early in cases with bulbar onset [4,8]. These manifestations may result from the involvement of cardiovascular and autonomic centers in the medulla oblongata. Pathological evidence has supported bulbar abnormalities in pTDP-43-related pathologies [26], possibly due to disease spread through contiguous anatomical structures rather than trans-synaptic propagation. This evidence might explain the co-occurrence of ALS–TTS, as previously described in several reports [10,11,12,19,20].

The relations between ALS and TTS and, more broadly, autonomic involvement in ALS have not been clearly understood and the currently available literature on this topic is heterogeneous and mostly based on case reports or small case series. In a Korean cohort study of 624 ALS patients, among 64 patients who underwent echocardiography after the presentation of cardiologic symptoms, TTS was detected in 9 cases, 8 of whom presented with bulbar or cervical onset and 1 with a respiratory onset, suggesting a potential role of bulbar involvement in TTS development [20]. Another retrospective analysis of a cohort of 250 ALS cases and 870 synucleinopathies found 4 TTS cases in the ALS group and none in the synucleinopathy group [13]. Interestingly, while Parkinson’s disease, Lewy body dementia, and multiple system atrophy are known for their association with autonomic dysfunction [27], their co-occurrence with TTS seems rather uncommon, indicating a specific link between ALS-specific pathological mechanisms and cardiomyopathy development [24].

Several hypotheses on TTS pathophysiology have been proposed, involving neurochemical, hormonal, and genetic factors that point towards a stress-triggered catecholaminergic toxicity [14]. TTS has been observed not only in the context of pathologies compromising the medulla oblongata but also in several acute neurological conditions without clear evidence of bulbar involvement, such as traumatic brain or spinal injury, cerebral hemorrhages, stroke, seizures, cerebral surgery, neuromuscular diseases, and psychiatric comorbidities such as anxiety and depression [24,28,29]. This supports the etiopathogenetic hypothesis of an underlying stress-triggered mechanism causing an imbalance in plasma levels of epinephrine, norepinephrine, and cortisol and subsequent sympathetic dysfunction at the level of cardiac myocytes.

Autonomic disturbances related to bulbar involvement, necessitating tracheostomy, tracheal intubation, or long-term respiratory support, may exacerbate the physiological stress response, promoting the onset of TTS or even favoring TTS recurrence [23]. ALS-related autonomic dysfunctions have also been assessed in the context of cardiovascular involvement, with studies showing altered HRV related to impaired vagal response in resting conditions, particularly in patients with prominent bulbar signs [5]. Altered responses to orthostasis have also been demonstrated in ALS through spectral analysis of HRV and systolic arterial pressure [6], as well as through cardiological assessments with tilting-tests, transthoracic echocardiography, and Holter-ECG [21].

Sympathetic hyperfunctioning has been demonstrated even in early stages of ALS. A retrospective analysis showed a significant increase in QTc intervals and dispersion at terminal stages, inversely correlated with a decrease in neuronal density in the sympathetic intermediolateral nucleus of the upper thoracic cord, also correlating with an increased rate of sudden cardiac arrest [25]. A heart MRI study on 35 ALS patients without overt cardiac involvement showed a reduction of myocardial mass and volumes in 77% of patients and myocardial fibrosis in 23.5%, with these structural changes hypothesized to be caused by sympathetic hyperactivation secondary to denervation of autonomic cardiac nerves or respiratory weakness [30]. Moreover, a 123-I-MIBG-SPECT study on early-stage ALS demonstrated a significantly reduced MIBG uptake in the majority of subjects, associated with a reduction in HRV, supporting an early sympathetic denervation process as a possible cause of cardiac dysfunction [9]. Loss of adrenergic receptors due to myocardial denervation has been linked to an increased responsiveness of the remaining receptors, supporting the idea that compensatory noradrenergic hyperfunction due to sympathetic denervation and ventilatory deficits, frequently observed in the context of bulbar involvement, could play a synergistic role in the development of TTS cardiomyopathy [31]. As the bulbar phenotype is characterized by a more aggressive disease progression, the stressful conditions secondary to disability progression may also contribute to sympathetic surge and serve as triggering factors for TTS [5,9].

## 4. Conclusions

TTS is a rare stress-related cardiomyopathy that may arise in the context of ALS. Sympathetic hyperfunction may play a crucial role in its pathogenesis and may be related to bulbar structures involvement, independently of onset type. Bulbar onset seems to relate to an earlier TTS presentation and to a more severe cardiological prognosis. However, the relation between ALS and TTS, as well as its relationship with bulbar involvement, needs further investigations in order to reveal the potential underlying pathological mechanisms.

## Figures and Tables

**Figure 1 ijms-24-12096-f001:**
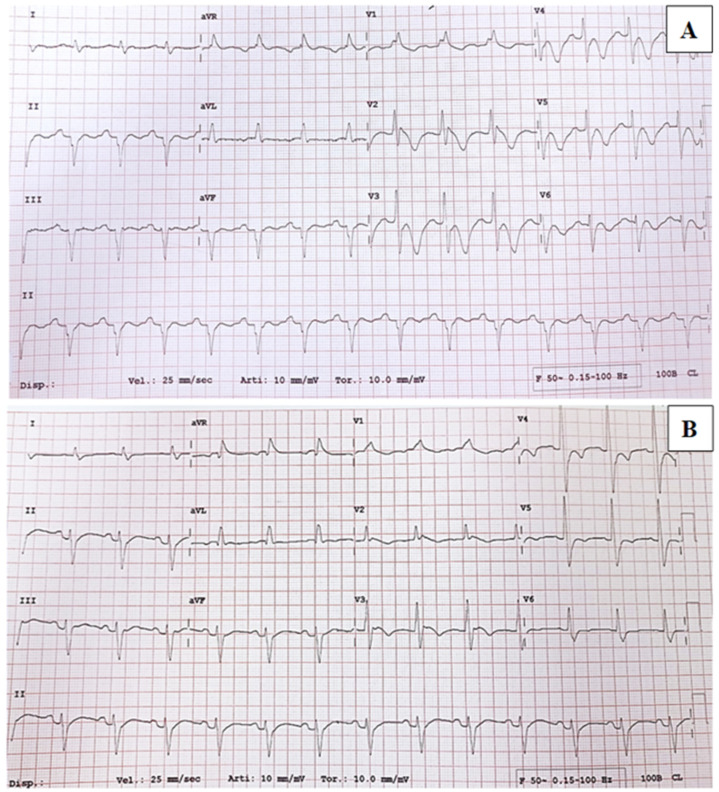
(**A**) ECG on admission revealed T-wave inversion on almost all leads, especially on precordial leads V2–V6, mimicking myocardial ischaemia. (**B**) ECG after 4 days revealed negative T-wave in anterior precordial leads, of smaller depth than previous one.

**Table 1 ijms-24-12096-t001:** Summary of the patient with ALS–TTS identified in our center. Abbreviations: ALS = amyotrophic lateral sclerosis; ALSFSR-R = Revised Amyotrophic Lateral Sclerosis Functional Rating Scale; CAD = coronary artery disease; CPK = creatine phosphokinase; HT = hypertension; LMN = lower motor neuron; M = male; MiToS = MiToS functional staging system; PE = pulmonary embolism; PEG = percutaneous endoscopic gastrostomy; PSVT = paroxysmal supraventricular tachycardia; T2DM = type 2 diabetes mellitus; TS = tracheostomy; TTS = Tako-Tsubo syndrome; UTI = urinary tract infection; † = exitus; * at TTS diagnosis.

Patient (Sex)	1 (Case Report, M)	2 (F)	3 (M)	4 (F)
ALS age of onset (years)	54	74	60	71
Site of onset	Upper limbs	Bulbar	Lower limbs	Lower limbs
ALS phenotype (according to Chiò, et al. 2011 [2])	Classic	Bulbar	LMN	Classic
ALS disease duration (from clinical onset)	73 months	19 months (†)	103 months (†)	38 months (†)
TTS (months after ALS onset)	73 months	11 months	86 months	38 months
Awaji at ALS diagnosis	Probable	Probable	Possible	Possible
ALSFRS-R *	15	45	19	18
KSS *	4	1	4	4
MiToS *	3	1	3	3
ALS genetics	/	C9orf72	/	/
Concomitant Medication	Riluzole, Acetyl-L-carnitine	Riluzole, Acetyl-L-carnitine, Baclofen	Riluzole	Riluzole
Time to Ventilation	73 months	N/A	86 months	32 months
Time to Tracheostomy	N/A	N/A	91 months	38 months
Time to PEG	75 months	N/A	N/A	38 months
Elevated ST/negative T at ECG	Prolonged QT, unspecific T wave alterations	Sinus tachycardia, VEBs	Negative antero-septal T	Aspecific conduction alterations
Troponin T (ng/dL)	231	421	459	316
pro-BNP (pg/mL)	939	34.309	3.074	/
CPK (U/L)	N/A	44	71	49
Ejection Fraction	45%	30%	35%	35%
Apical dyskinesia/ballooning at echocardiography	Yes	Yes	Yes	Yes
Coronary CT/coronarography	Heart angio-CT: negative	Coronarography: negative	N/A	Heart CT: negative
Pulmonary arterial Pressure	Normal	75 mmHg	Normal	Normal
Cardiovascular comorbidities	PSVT	HT	HT	T2DM, CAD
Intervening condition	Respiratory failure (adapted to NIV)	PE	PE + lower UTI	TS + PEG

**Table 2 ijms-24-12096-t002:** Summary of the ALS–TTS reported cases in the literature, focusing on bulbar impairment as TTS possible cause. Abbreviations: CR = case report; CS = case series; NIV = non-invasive ventilation; TIV = tracheostomy invasive ventilation; UTI = urinary tract infection.

Reference	Number of Cases	Type of Onset	Ventilatory Support	Bulbar Signs	Intervening Conditions
Takayama et al., 2004 [12]	1	Classic	N/A	N/A	Surgical gastrostomy; repair of incisional hernia (with tracheal intubation)
Mitani et al., 2005 [11]	1	Bulbar	NIV	Dysphagia, dysarthria.	Long-term NIV
Matsuyama et al., 2008 [10]	1	Classic	TIV	N/A	Tracheostomy
Massari et al., 2011 [19]	1	Classic	NIV	Dysphagia, dysarthria	Pneumonia
Peters, 2014 [22]	1	N/A	NIV	N/A	Respiratory distress syndrome, pneumonia
Santoro et al., 2016 [23]	1	N/A	N/A	N/A	Emotional distress (first episode); Femoral artery thrombosis (second episode)
Gdynia et al., 2006 [24]	1	Classic	N/A	N/A	Major surgery
Choi et al., 2017 [20]	9	Bulbar (pt.2,5,6) Cervical (pt.1,3,4,7,8) Respiratory (pt.9)	NIV (pt.2,5,7), TIV (pt.6,8)	N/A	N/A
Izumi et al., 2018 [13]	4	Bulbar (pt.1,4), Classic (pt.2,3)	TIV (pt.1,2)	N/A	UTI (pt.1); acute cholangitis (pt.2); pneumonia (pt.3)

## Data Availability

All data needed to evaluate the conclusions are present in the paper. Additional data related to this paper may be requested from the corresponding author upon reasonable request by qualified academic investigators.

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
