# Peer review of "Tako-Tsubo Syndrome in Amyotrophic Lateral Sclerosis: Single-Center Case Series and Brief Literature Review"

_ijms, 2023, doi:10.3390/ijms241512096_

Round 1
Reviewer 1 Report
An original view of TTS in ALS. I have some comments:
1.EES criteria were used - why not more recent Awaji criteria were not used?
2.Choi - 9 bulbar form patients of ALS with TTS. This findings is very suspicious.
3.ALS and dysautonomia - this is not in relation with TTS? This not very clearly presented for readers.
4.Catecholamine toxicity and development of TTS. I suggest to use some situation of TTS in neurology - some strokes, depressions, etc. And the findings compare to Your ALS-TTS patients.
5.Only bulbar symptoms in ALS are associated with TTS? And other neurologic diseases without bulbar syndrome are associated with TTS. This has to be more profoundly discussed.
6.30 referencies - only 3 are really recent.
No comments
Author Response
Point-by-point response
Reviewer 1:
- “EES criteria were used - why not more recent Awaji criteria were not used?”
Reply:
We thank the reviewer for this suggestion. We initially used the El Escorial criteria (EEC) because some of our patients were diagnosed almost 10 years ago and were initially evaluated using ECC. However, we acknowledge the reviewer's point that the Awaji criteria are more recent and have now reported the Awajii criteria in Table 1.
- “Choi - 9 bulbar form patients of ALS with TTS. This findings is very suspicious.”
Reply:
We thank the reviewer for pointing out this issue. We double checked the clinical data reported in the paper and have now checked and updated Table 2. We also checker the Discussion accordingly.
- “ALS and dysautonomia - this is not in relation with TTS? This not very clearly presented for readers.”
Reply:
Yes, in order to clarify this issue, in the Introduction we implemented a general overview on Tako-Tsubo syndrome (TTS), stating that it is triggered by adrenergic dysregulation, providing insight into the possible connection between ALS, dysautonomia, and TTS.
Moreover, in the Discussion, we further explored this connection in the following paragraphs, to provide a clearer link between the two pathologies:
- On lines 133-139, we reported various dysautonomic dysfunctions observed in ALS, including parasympathetic hypofunctioning and sympathetic hyperfunctioning [6,21,22]. These dysfunctions usually appear in late stages of ALS, but they have been rarely reported in cases with bulbar onset [4,8].
- On lines 139-143, we mentioned pathological evidence of cardiac centers involvement in ALS, possibly explaining the co-occurrence of ALS-TTS. This evidence shows bulbar abnormalities in pTDP-43-related pathologies [23], indicating potential disease spread through contiguous anatomical structures rather than trans-synaptic propagation. This could also explain the association of ALS-TTS, as observed in several reports [10-12,19,20,24].
- “Catecholamine toxicity and development of TTS. I suggest to use some situation of TTS in neurology - some strokes, depressions, etc. And the findings compare to Your ALS-TTS patients.”
- “Only bulbar symptoms in ALS are associated with TTS? And other neurologic diseases without bulbar syndrome are associated with TTS. This has to be more profoundly discussed.”
Reply:
In the Discussion section, we have made further changes, particularly regarding other neurological conditions associated with Tako-Tsubo syndrome (TTS) and the underlying pathophysiological mechanisms. However, direct comparison between our cases and previously described cases of TTS in a setting of neurological impairment remains challenging due to the limited clinical data provided by such case reports and series. We have also emphasized the concept that TTS may occur in pathologies that spare the bulbar functions, highlighting the role of catecholaminergic imbalance leading to sympathetic dysfunction at the level of cardiac myocytes. This provides valuable insights into the potential triggers of TTS in various neurological contexts beyond just bulbar involvement.
- “30 references - only 3 are really recent.”
We agree with the reviewer that a part of the references is older than 2010. The literature on this topic is heterogeneous mostly based on case reports or small case series. However, being the relationship between ALS and TTS a niche topic, there’s only a limited number of studies exploring this association. We tried to include work relevant to our Discussion, regardless of their publication year, trying to give to reader a grasp of the topic and of the available evidence. However, we are open to further suggestions if the reviewer know more relevant articles on this topic which could improve our work.
Reviewer 2 Report
The article proposed by Napoli and colleagues is certainly interesting. Indeed, it considers the correlation between Tako-Tsubo syndrome and amyotrophic lateral sclerosis, which is little known in the literature and deserves further investigation.
However, it is necessary to implement the introduction, describing the syndrome better to make the article accessible to a wider audience.
Minor considerations: always use the abbreviation 'TTS-ALS' or 'ALS-TTS' and not both; line 86 'ALS' instead of 'AL'.
Author Response
Point-by-point response
Reviewer 2:
- “However, it is necessary to implement the introduction, describing the syndrome better to make the article accessible to a wider audience.”
The introduction has been implemented as suggested by the reviewer; further information about TTS which are relevant for our purpose are provided later in the Discussion, highlighting common pathological mechanisms of ALS involving bulbar function and cardiac dysautonomia.
- “Minor considerations: always use the abbreviation 'TTS-ALS' or 'ALS-TTS' and not both; line 86 'ALS' instead of 'AL'.”
The errors have been corrected and the abbreviations are now consistent throughout the manuscript.